# Asynchronous Parallel Coordinate Minimization for MAP Inference

**Ofer Meshi**
Google
meshi@google.com

**Alexander G. Schwing**
Department of Electrical and Computer Engineering
University of Illinois at Urbana-Champaign
aschwing@illinois.edu

## Abstract

Finding the maximum a-posteriori (MAP) assignment is a central task for structured prediction. Since modern applications give rise to very large structured problem instances, there is increasing need for efficient solvers. In this work we propose to improve the efficiency of coordinate-minimization-based dual-decomposition solvers by running their updates asynchronously in parallel. In this case message-passing inference is performed by multiple processing units simultaneously without coordination, all reading and writing to shared memory. We analyze the convergence properties of the resulting algorithms and identify settings where speedup gains can be expected. Our numerical evaluations show that this approach indeed achieves significant speedups in common computer vision tasks.

## 1 Introduction

Finding the most probable configuration of a structured distribution is an important task in machine learning and related applications. It is also known as the maximum a-posteriori (MAP) inference problem in graphical models [Wainwright and Jordan, 2008, Koller and Friedman, 2009], and has found use in a wide range of applications, from disparity map estimation in computer vision, to part-of-speech tagging in natural language processing, protein-folding in computational biology and others. Generally, MAP inference is intractable, and efficient algorithms only exist in some special cases, such as tree-structured graphs. It is therefore common to use approximations.

In recent years, many approximate MAP inference methods have been proposed [see Kappes et al., 2015, for a recent survey]. One of the major challenges in applying approximate inference techniques is that modern applications give rise to very large instances. For example, in semantic image segmentation the task is to assign labels to all pixels in an image [*e.g.*, Zhou et al., 2016]. This can translate into a MAP inference problem with hundreds of thousands of variables (one for each pixel). For this reason, efficiency of approximate inference algorithms is becoming increasingly important.

One approach to dealing with the growth in problem complexity is to use cheap (but often inaccurate) algorithms. For example, variants of the mean field algorithm have witnessed a surge in popularity due to their impressive success in several computer vision tasks [Krähenbühl and Koltun, 2011]. A shortcoming of this approach is that it is limited to a specific type of model (fully connected graphs with Gaussian pairwise potentials). Moreover, the mean field approximation is often less accurate than other approximations, *e.g.*, those based on convex relaxations [Desmaison et al., 2016].

In this work we study an alternative approach to making approximate MAP inference algorithms more efficient – *parallel computation*. Our study is motivated by two developments. First, current hardware trends increase the availability of parallel processing hardware in the form of multi-core CPUs as well as GPUs. Second, recent theoretical results improve our understanding of various asynchronous parallel algorithms, and demonstrate their potential usefulness, especially for objective functions that are typical in machine learning problems [*e.g.*, Recht et al., 2011, Liu et al., 2015].

Focusing on a smoothed objective function originating from a dual-decomposition approximation, we present a fully asynchronous parallel algorithm for MAP inference based on block-coordinate updates. Our approach gives rise to a message-passing procedure, where messages are computed and updated in shared memory asynchronously in parallel by multiple processing units, with no attempt to coordinate their actions. The reason we focus on asynchronous algorithms is because the runtime of synchronous algorithms is dominated by the slowest worker, which may cause the overhead from synchronization to outweigh the gain from parallelization. The asynchronous parallel setting is particularly suitable for message-passing algorithms, like the ones we study here.

Our analysis is conducted under the *bounded delay* assumption, which is standard in the literature on asynchronous optimization and matches well modern multicore architectures. It reveals the precise relation between the delay and the expected change in objective value following an update. This result suggests a natural criterion for adaptively choosing the number of parallel workers to guarantee convergence to the optimal value. Additional analysis shows that speedups which are linear in the number of processors can be expected under some conditions. We illustrate the performance of our algorithm both on synthetic models and on a disparity estimation task from computer vision. We demonstrate 45-fold improvements or more when compared to other asynchronous optimization techniques.

## 2  Related Work

Our work is inspired by recent advances in the study of asynchronous parallel algorithms and their successful application to various machine learning tasks. In particular, parallel versions of various sequential algorithms have been recently analyzed, adding to past work in asynchronous parallel optimization [Bertsekas and Tsitsiklis, 1989, Tseng, 1991]. Those include, for example, stochastic gradient descent [Recht et al., 2011], conditional gradient [Wang et al., 2016], ADMM [Zhang and Kwok, 2014], proximal gradient methods [Davis et al., 2016], and coordinate descent [Liu et al., 2015, Liu and Wright, 2015, Avron et al., 2015, Hsieh et al., 2015, Peng et al., 2016, You et al., 2016].

The algorithms we study here are based on *block coordinate minimization*, a coordinate descent method in which an *optimal* update is computed in closed form.[1] To the best of our knowledge, this algorithm has yet to be analyzed in the asynchronous parallel setting. The analysis of this algorithm is significantly more challenging compared to other coordinate descent methods, since there is no notion of a step-size, which is carefully chosen in previous analyses to guarantee convergence [*e.g.*, Liu et al., 2015, Avron et al., 2015, Peng et al., 2016]. Furthermore, in most previous papers, the function which is being optimized is assumed to be strongly convex, or to satisfy a slightly weaker condition [Liu et al., 2015, Hsieh et al., 2015]. In contrast, we analyze a smooth and convex MAP objective, which does not satisfy any of these strong-convexity conditions. We focus on this particular objective function since optimal block coordinate updates are known in this case, which is not true for its strongly convex counterparts [Meshi et al., 2015].

We are not the first to study parallel inference methods in graphical models. Parallel variants of Belief Propagation (BP) are proposed and analyzed by Gonzalez et al. [2011]. They present bounds on achievable gains from parallel inference on chain graphs, as well as an optimal parallelization scheme. However, the algorithms they propose include global synchronization steps, which often hurt efficiency. In contrast, we focus on the fully asynchronous setting, so our algorithms and analysis are substantially different. Piatkowski and Morik [2011] and Ma et al. [2011] also describe parallel implementations of BP, but those again involve synchronization. We are particularly interested in the MAP inference problem and use convergent coordinate minimization methods with a dual-decomposition objective. This is quite different from marginal inference with BP, used in the aforementioned works; for example, BP is not guaranteed to converge even with sequential execution.

Dual-decomposition based parallel inference for graphical models has been investigated by Choi and Rutenbar [2012] and extended by Hurkat et al. [2015]. They study hardware implementations of the TRW-S algorithm (a coordinate-minimization algorithm very similar to the ones we study here), where some message computations can be parallelized. However, their parallelization scheme is quite different from ours as it is synchronous, *i.e.*, the messages computed in parallel have to be carefully chosen, and it is specific to grid-structured graphs. In addition, they provide no theoretical analysis

of convergence (which is not directly implied by TRW-S convergence due to different message scheduling).

Schwing et al. [2011] and Zhang et al. [2014] also study dual-decomposition based parallel inference. They demonstrate gains when parallelizing the computation across multiple machines in a cluster. However, their approach requires the employed processing units to run in synchrony. Parallel MAP solvers based on subdifferential techniques [Schwing et al., 2012], have also been considered by Schwing et al. [2014] using a Frank-Wolfe algorithm. Albeit individual computations are performed in parallel, their approach also requires a synchronous gradient step.

An alternative parallel inference approach is based on sampling algorithms [Singh et al., 2010, Wick et al., 2010, Asuncion et al., 2011]. However, the gains in runtime observed in this case are usually much smaller than those observed for algorithms which do not use sampling.

Our work is thus the first to propose and analyze a fully asynchronous parallel coordinate minimization algorithm for MAP inference in graphical models.

## 3  Approach

In this section we formalize the MAP inference problem and present our algorithmic framework. Consider a set of discrete variables $X_1, \ldots, X_N$, and denote by $x_i \in \mathcal{X}_i$ a particular assignment to variable $X_i$ from a discrete set $\mathcal{X}_i$. Let $r \subseteq \{1, \ldots, N\}$ denote a subset of the variables, also known as a *region*, and let $\mathcal{R}$ be the set of all regions that are used in a problem. Each region $r \in \mathcal{R}$ is associated with a local score function $\theta_r(x_r)$, referred to as a *factor*. The MAP inference problem is to find a joint assignment $x$ that maximizes the sum of all factor scores,

$$\max_x \sum_{r \in \mathcal{R}} \theta_r(x_r) \; . \tag{1}$$

Consider semantic image segmentation as an example. Factors depending on a single variable denote univariate preferences often obtained from neural networks [Chen* et al., 2015]. Factors depending on two or more variables encode local preference relationships.

The problem in Eq. (1) is a combinatorial optimization problem which is generally NP-hard [Shimony, 1994]. Notable tractable special cases include tree-structured graphs and super-modular pairwise factors. In this work we are interested in solving the general form of the problem, therefore we resort to approximate inference.

Multiple ways to compute an approximate MAP solution have been proposed. Here we employ approximations based on the *dual-decomposition* method [Komodakis et al., 2007, Werner, 2010, Sontag et al., 2011], which often deliver competitive performance compared to other approaches, and are also amenable to asynchronous parallel execution. The key idea in dual-decomposition is to break the global optimization problem of Eq. (1) into multiple (easy) subproblems, one for each factor. Agreement constraints between overlapping subproblem maximizers are then defined, and the resulting program takes the following form,[2]

$$\min_{\delta} \sum_{r \in \mathcal{R}} \max_{x_r} \left( \theta_r(x_r) + \sum_{p:r \in p} \delta_{pr}(x_r) - \sum_{c:c \in r} \delta_{rc}(x_c) \right) \; \equiv \; \min_{\delta} \sum_{r \in \mathcal{R}} \max_{x_r} \hat{\theta}_r^{\delta}(x_r) \; . \tag{2}$$

Here, '$r \in p$' (similarly, '$c \in r$') represents parent-child containment relationships, often represented as a region graph [Wainwright and Jordan, 2008], and $\delta$ are Lagrange multipliers for the agreement constraints, defined for every region $r$, assignment $x_r$, and parent $p : r \in p$. In particular, these constraints enforce that the maximizing assignment in a parent region $p$ agrees with the maximizing assignment in the child region $r$ on the values of the variables in $r$ (which are also in $p$ due to containment). For a full derivation see Werner [2010] (Eq. (11)). The modification of the model factors $\theta_r$ by the multipliers $\delta$ is known as a *reparameterization*, and is denoted here by $\hat{\theta}_r^{\delta}$ for brevity.

The program in Eq. (2) is an unconstrained convex problem with a (piecewise-linear) non-smooth objective function. Standard algorithms, such as subgradient descent, can be applied in this case [Komodakis et al., 2007, Sontag et al., 2011], however, often, faster algorithms can be derived for a smoothed variant of this objective function [Johnson, 2008, Hazan and Shashua, 2010, Werner, 2009,

**Algorithm 1** Block Coordinate Minimization
---
1: Initialize: $\delta^0 = 0$
2: **while** not converged **do**
3:     Choose a block $s$ at random
4:     Update: $\delta_s^{t+1} = \operatorname{argmin}_{\delta_s'} f(\delta_s', \delta_{-s}^t)$,     and keep: $\delta_{-s}^{t+1} = \delta_{-s}^t$
5: **end while**
---

Savchynskyy et al., 2011]. In this approach the max operator is replaced with *soft-max*, giving rise to the following problem:

$$\min_\delta f(\delta) := \sum_{r \in \mathcal{R}} \gamma \log \sum_{x_r} \exp\left(\hat{\theta}_r^\delta(x_r)/\gamma\right) , \tag{3}$$

where $\gamma$ is a parameter controlling the amount of smoothing (larger is smoother).

**Algorithms:** Several algorithms for optimizing either the smooth (Eq. (3)) or non-smooth (Eq. (2)) problem have been studied. *Block coordinate minimization* algorithms, which are the focus of our work, are among the most competitive methods. In particular, in this approach a block of variables $\delta_s$ is updated at each iteration using the values in other blocks, *i.e.*, $\delta_{-s}$, which are held fixed. Below we will assume a randomized schedule, where the next block to update is chosen uniformly at random. Other schedules are possible [*e.g.*, Meshi et al., 2014, You et al., 2016], but this one will help to avoid unwanted coordination between workers in an asynchronous implementation. The resulting meta-algorithm is given in Algorithm 1.

Various choices of blocks give rise to different algorithms in this family. A key consideration is to make sure that the update in line 4 of Algorithm 1 can be computed efficiently. Indeed, for several types of blocks, efficient, oftentimes analytically computable, updates are known [Werner, 2007, Globerson and Jaakkola, 2008, Kolmogorov, 2006, Sontag et al., 2011, Meshi et al., 2014]. To make the discussion concrete, we next instantiate the block coordinate minimization update (line 4 in Algorithm 1) using the smooth objective in Eq. (3) for two types of blocks.[3] Specifically, we use the *Pencil block*, consisting of the variables $\delta_{pr}(\cdot)$, and the *Star block*, which consists of the set $\delta_{\cdot r}(\cdot)$. Intuitively, for the Pencil block, we choose a parent $p$ and one of its children $r$. For the Star block we choose a region $r$ and consider all of its parents.

To simplify notation, it is useful to define per-factor probability distributions, referred to as *beliefs*:

$$\mu_r(x_r) \propto \exp\left(\hat{\theta}_r^\delta(x_r)/\gamma\right) .$$

Using this definition, the Pencil update is performed by picking a pair of adjacent regions $p, r$, and setting:

$$\delta_{pr}^{t+1}(x_r) = \delta_{pr}^t(x_r) + \frac{1}{2}\gamma\left(\log \mu_p^t(x_r) - \log \mu_r^t(x_r)\right) \tag{4}$$

for all $x_r$, where we denote the marginal belief $\mu_p(x_r) = \sum_{x_{p\backslash r}'} \mu_p(x_r, x_{p\backslash r}')$. Similarly, for the Star update we pick a region $r$, and set:

$$\delta_{pr}^{t+1}(x_r) = \delta_{pr}^t(x_r) + \gamma \log \mu_p^t(x_r) - \frac{1}{P_r + 1} \cdot \gamma \log \left(\mu_r^t(x_r) \cdot \prod_{p':r\in p'} \mu_{p'}^t(x_r)\right)$$

for all $p : r \in p$ and all $x_r$, where $P_r = |\{p : r \in p\}|$ is the number of parents of $r$ in the region graph. Full derivation of the above updates is outside the scope of this paper and can be found in previous work [e.g., Meshi et al., 2014]. The variables $\delta$ are sometimes called *messages*. Hence the algorithms considered here belong to the family of message-passing procedures.

In terms of convergence rate, it is known that coordinate minimization converges to the optimum of the smooth problem in Eq. (3) with rate $O(1/\gamma t)$ [Meshi et al., 2014].

In this work our goal is to study asynchronous parallel coordinate minimization for approximate MAP inference. This means that each processing unit repeatedly performs the operations in lines 3-4

of Algorithm 1 independently, with minimal coordination between units. We refer to this algorithm as *APCM* – for Asynchronous Parallel Coordinate Minimization. We use *APCM-Pencil* and *APCM-Star* to refer to the instantiations of APCM with Pencil and Star blocks, respectively.

## 4   Analysis

We now proceed to analyze the convergence properties of the asynchronous variants of Algorithm 1. In this setting, the iteration counter $t$ corresponds to write operations, which are assumed to be atomic. Note, however, that in our experiments in Section 5 we use a lock-free implementation, which may result in inconsistent writes and reads.

If there is no delay, then the algorithm is performing exact coordinate minimization. However, since updates happen asynchronously, there will generally be a difference between the current beliefs $\mu^t$ and the ones used to compute the update. We denote by $k(t)$ the iteration counter corresponding to the time in which values were read. The bounded delay assumption implies that $t - k(t) \leq \tau$ for some constant $\tau$. We present results for the Pencil block next, and defer results for the Star block to Appendix B.

Our first result precisely characterizes the expected change in objective value following an update as a function of the old and new beliefs. All proofs appear in the supplementary material.

**Proposition 1.** *The APCM-Pencil algorithm satisfies:*

$$\mathbb{E}_s[f(\delta^{t+1})] - f(\delta^t) = \frac{\gamma}{n} \sum_r \sum_{p:r \in p} \left( \log \sum_{x_r} \frac{\mu_r^t(x_r)}{\mu_r^{k(t)}(x_r)} \sqrt{\mu_p^{k(t)}(x_r) \cdot \mu_r^{k(t)}(x_r)} \right. \tag{5}$$
$$\left. + \log \sum_{x_r} \frac{\mu_p^t(x_r)}{\mu_p^{k(t)}(x_r)} \sqrt{\mu_p^{k(t)}(x_r) \cdot \mu_r^{k(t)}(x_r)} \right),$$

*where $n = \sum_r \sum_{p:r \in p} 1$ is the number of Pencil blocks, and the expectation is over the choice of blocks.*

At a high-level, our derivation carefully tracks the effect of stale beliefs on convergence by separating old and new beliefs after applying the update (see Appendix A.1). We next highlight a few consequences of Proposition 1. First, it provides an *exact* characterization of the expected change in objective value, not an upper bound. Second, as a sanity check, when there is no delay ($k(t) = t$), the belief ratio terms ($\mu^t/\mu^{k(t)}$) drop, and we recover the sequential decrease in objective, which corresponds to the (negative) Bhattacharyya divergence measure between the pair of distributions $\mu_r^t(x_r)$ and $\mu_p^t(x_r)$ [Meshi et al., 2014]. Finally, Proposition 1 can be used to dynamically set the degree of parallelization as follows. We estimate Eq. (5) (per block) and if the result is strictly positive then it suggests that the delay is too large and we should reduce the number of concurrent processors.

Next, we obtain an upper bound on the expected change in objective value that takes into account the sparsity of the update.

**Proposition 2.** *The APCM-Pencil algorithm satisfies:*

$$\mathbb{E}_s[f(\delta^{t+1})] - f(\delta^t) \leq \frac{\gamma}{n} \sum_{d=k(t)}^{t-1} \left[ \max_{x_r} \left( \log \frac{\mu_{r(d)}^{d+1}(x_r)}{\mu_{r(d)}^d(x_r)} \right) + \max_{x_r} \left( \log \frac{\mu_{p(d)}^{d+1}(x_r)}{\mu_{p(d)}^d(x_r)} \right) \right] \tag{6}$$

$$+ \frac{\gamma}{n} \sum_r \sum_{p:r \in p} \log \left( \sum_{x_r} \sqrt{\mu_p^{k(t)}(x_r) \cdot \mu_r^{k(t)}(x_r)} \right)^2. \tag{7}$$

This bound separates the expected change in objective into two terms: the delay term (Eq. (6)) and the (stale) improvement term (Eq. (7)). The improvement term is always non-positive, it is equal to the negative Bhattacharyya divergence, and it is exactly the same as the expected improvement in the sequential setting. The delay term is always non-negative, and as before, when there is no delay ($k(t) = t$), the sum in Eq. (6) is empty, and we recover the sequential improvement. Note that the delay term depends only on the beliefs in regions that were actually updated between the read and current write. This result is obtained by exploiting the sparsity of the updates: each message affects only the neighboring nodes in the graph (see Appendix A.2). Similar structural properties are also used in related analyses [*e.g.*, Recht et al., 2011], however in other settings this involves making

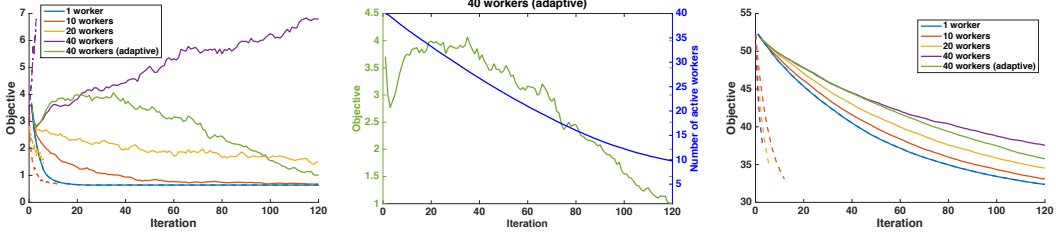

Figure 1: Simulation of APCM-Pencil on toy models. (Left) objective vs. iteration (equiv., update) on a 3-node chain graph. The dashed lines show the same objective when iterations are divided by the number of workers, which approximates runtime. (Middle) objective vs. iteration and vs. number of active workers on a 3-node chain graph when adapting the number of workers. (Right) objective vs. iteration (equiv., update) on a 6-node fully connected graph.

non-trivial assumptions (such as how training examples interact), whereas in our case the sparsity pattern is readily available through the structure of the graphical model.

To demonstrate the hardness of our setting, we present in Appendix A.3 a case where the RHS of Eq. (6) - (7) may be a large positive number. This happens when some beliefs are very close to 0. In contrast, the next theorem uses the results above to show speedups under additional assumptions.

**Theorem 1.** *Let $|\hat{\theta}_r^{\delta^t}(x_r)| \leq M$ for all $t, r, x_r$, and let $\|\delta^t - \delta^*\|^2 < B$ for all t. Assume that the gradient is bounded from below as $\|\nabla f\|^2 \geq c$, and that the delay is bounded as $\tau \leq \frac{\gamma c}{32M}$. Then $\mathbb{E}_s[f(\delta^t)] - f(\delta^*) \leq \frac{8nB}{\gamma t}$.*

This upper bound is only 2 times slower than the corresponding sequential bound (see Theorem 3 in Meshi et al. [2014]), however, in this parallel setting we execute updates roughly $\tau$ times faster, so we obtain a linear speedup in this case. Notice that this rate applies only when the gradient is not too small, so we expect to get large gains from parallelization initially, and smaller gains as we get closer to optimality. This is due to the hardness of our setting (see Appendix A.3), and gives another theoretical justification to adaptively reduce the number of processing units as the iterations progress.

At first glance, the assumptions in Theorem 1 (specifically, the bounds $M$ and $B$) seem strong. However, it turns out that they are easily satisfied whenever $f(\delta^t) \leq f(0)$ (see Lemma 9 in Meshi et al. [2014]) – which is a mild assumption that is satisfied in all of our experiments except some adversarially constructed toy problems (see Section 5.1).

## 5 Experiments

In this section we present numerical experiments to study the performance of APCM in practical MAP estimation problems. We first simulate APCM on toy problems in Section 5.1, then, in Section 5.2, we demonstrate our approach on a disparity estimation task from computer vision.

### 5.1 Synthetic Problems

To better understand the behavior of APCM, we simulate the APCM-Pencil algorithm sequentially as follows. We keep a set of 'workers,' each of which can be in one of two states: 'read' or 'update.' In every step, we choose one of the workers at random using a skewed distribution to encourage large delays: the probability of sampling a worker $w$ is $p_w = e^{\kappa s_w}/\sum_{w'} e^{\kappa s_{w'}}$, where $s_w$ is sampled uniformly in $[0, 1]$, and $\kappa = 5$. If the worker is in the 'read' state, then it picks a message uniformly at random, makes a local copy of the beliefs, and moves to state 'update.' Else, if the worker wakes up in state 'update,' then it computes the update from its local beliefs, writes the update to the global beliefs, and goes back to state 'read.' This procedure creates delays between the read and write steps.

Our first toy model consists of 3 binary variables and 2 pairwise factors, forming a chain graph. This model has a total of 4 messages. Factor values are sampled uniformly in the range $[-5, 5]$. In Fig. 1 (left) we observe that as the number of workers grows, the updates become less effective due to stale beliefs. Importantly, it takes 40 workers operating on 4 messages to observe divergence. We don't

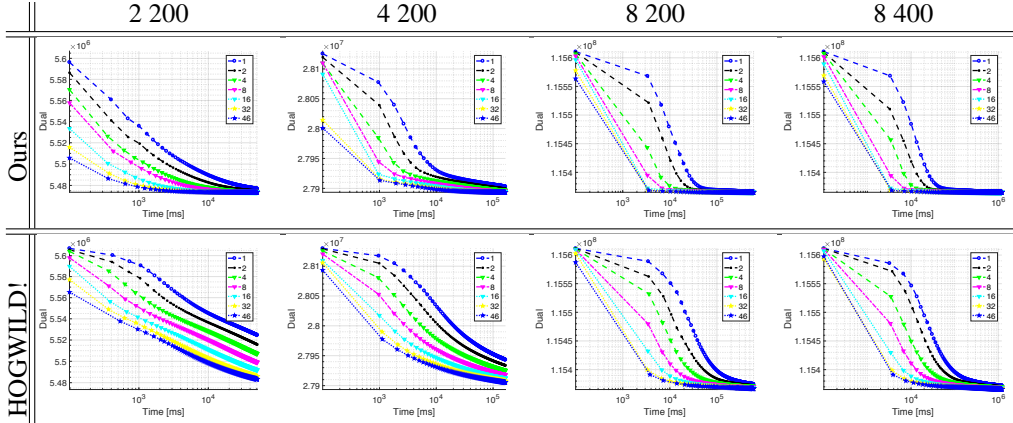

Figure 2: For $\gamma = 1$ and an 8 state model, we illustrate the convergence behavior of our approach compared to HOGWILD!, for a variety of MRF configurations (2, 4, 8), and different number of iterations (200, 400). Different number of threads are used for each configuration.

---

**Algorithm 2** HOGWILD! A single update

1: Choose a region $r \in \mathcal{R}$ at random
2: Update: $\delta_{pr}(x_r) \mathrel{-}= \eta_t \mu_r(x_r)$ for all $x_r, p : r \in p$
$\qquad\qquad \delta_{rc}(x_c) \mathrel{+}= \eta_t \mu_r(x_c)$ for all $x_c, c : c \in r$

---

expect a setting with more workers than messages to be observed in practice. We also adaptively change the number of workers as suggested by our theory, which indeed helps to regain convergence. Fig. 1 (middle) shows how the number of workers decreases as the objective approaches the optimum.

Our second toy model consists of 6 binary variables forming a fully connected graph. This model has 30 messages. In this setting, despite stale beliefs due to a skewed distribution, Fig. 1 (right) shows that APCM is convergent even with 40 active workers. Hypothetically assuming 40 workers to run in parallel yields a significant speedup when compared to a single thread, as is illustrated by the dashed lines in Fig. 1.

### 5.2 Disparity Estimation

We now proceed to test our approach on a disparity estimation task, a more realistic setup. In our case, the employed pairwise graphical model, often also referred to as a pairwise Markov random field (MRF), is grid structured. It has $144 \times 185 = 26,640$ unary regions with 8 states and is a downsampled version from Schwing et al. [2011]. We use the temperature parameter $\gamma = 1$ for the smooth objective (Eq. (3)). We compare our APCM-Star algorithm to the HOGWILD! approach [Recht et al., 2011], which employs an asynchronous parallel stochastic gradient descent method – summarized in Algorithm 2, where we use the shorthand $\mu_r(x_c) = \sum_{x'_{r \backslash c}} \mu_r(x_c, x'_{r \backslash c})$. We refer the reader to Appendix C in the supplementary material for additional results on graphical models with larger state space size and for results regarding the non-smooth update obtained for $\gamma = 0$. In short, those results are similar to the ones reported here.

No synchronization is used for both HOGWILD! and our approach, *i.e.*, we allow inconsistent reads and writes. Hence our optimization is lock-free and each of the threads is entirely devoted to computing and updating messages. We use one additional thread that constantly monitors progress by computing the objective in Eq. (3). We perform this function evaluation a fixed number of times, either 200 or 400 times. Running for more iterations lets us compare performance in the high-accuracy regime. During function evaluation, other threads randomly and independently choose a region $r$ and update the variables $\delta_{\cdot r}(\cdot)$, *i.e.*, we evaluate the Star block updates of Eq. (5). Our choice is motivated by the fact that Star block updates are more overlapping compared to Pencil updates, as they depend on more variables. Therefore, Star blocks are harder to parallelize (see Theorem 2 in Appendix B).

To assess the performance of our technique we use pairwise graphical models of different densities. In particular, we use a 'connection width' of 2, 4, or 8. This means we connect variables in the grid by

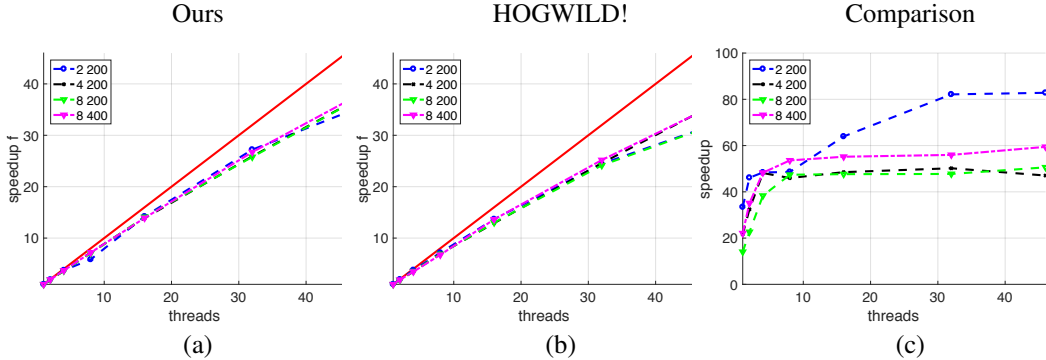

Figure 3: Speedup w.r.t. single thread obtained for a specific number of threads for our approach (a) and HOGWILD! (b), using a variety of MRF neighborhoods (2, 4, 8), and different number of iterations (200, 400). Speedups are shown for $\gamma = 1$ and 8 states. (c) shows the speedup of our method compared to HOGWILD!.

pairwise factors, if their $\ell_\infty$-norm distance is less than 2, 4, or 8. A 'connection width' of 2 is often also referred to as 8-neighborhood, because a random variable is connected to its eight immediate neighbors. A 'connection width' of 4 or 8 connects a random variable to 48 or 224 neighboring variables respectively. Hence, the connectivity of the employed graphical model is reasonably dense to observe inconsistent reads and writes. At the same time our experiments cover connection densities well above many typical graphical models used in practice.

**Convergence:** In a first experiment we investigate the convergence behavior of our approach and the HOGWILD! implementation for different graphical model configurations. We examine the behavior when using one to 46 threads, where the number of threads is not adapted, but remains fixed throughout the run. The stepsize parameter, necessary in the case of HOGWILD!, is chosen to be as large as possible while still ensuring convergence (following Recht et al. [2011]). Note that our approach is hyper-parameter free. Hence no tuning is required, which we consider an important practical advantage. We also evaluated HOGWILD! using a diminishing stepsize, but found those results to be weaker than the ones reported here. Also note that a diminishing stepsize introduces yet another hyper-parameter. Our results are provided in Fig. 2 for $\gamma = 1$ and 8 states per random variable. We assess different MRF configurations (2, 4, 8 connectivity), and iterations (200, 400). Irrespective of the chosen setup, we observe monotone convergence even with 46 threads at play for both approaches. In neither of our configurations do we observe any instability during optimization. As expected, we also observe the exact minimization employed in our approach to result in significantly faster descent than use of the gradient (*i.e.*, HOGWILD!). This is consistent with the comparison of these methods in the sequential setting.

**Thread speedup:** In our second experiment we investigate the speedup obtained when using an increasing number of threads. To this end we use the smallest dual value obtained with a single thread and illustrate how much faster we are able to obtain an identical or better value when using more than one thread during computation. The results for all the investigated graphical model configurations are illustrated in Fig. 3 (a) for our approach and in Fig. 3 (b) for HOGWILD!. In these figures, we observe very similar speedups across different graphical model configurations. We also observe that our approach scales just as well as the gradient based technique does.

**HOGWILD! speedup:** In our third experiment we directly compare HOGWILD! to our approach. More specifically, we use the smallest dual value found with the gradient based technique using a fixed number of threads, and assess how much faster the proposed approach is able to find an identical or better value when using the same number of threads. We show speedups of our approach compared to HOGWILD! in Fig. 3 (c). Considering the results presented in the previous paragraphs, speedups are to be expected. In all cases, we observe the speedups to be larger when using more threads. Depending on the model setup, we observe speedups to stabilize at values around 45 or higher.

In summary, we found our asynchronous optimization technique to be a compelling practical approach to infer approximate MAP configurations for graphical models.

## 6 Conclusion

We believe that parallel algorithms are essential for dealing with the scale of modern problem instances in graphical models. This has led us to present an asynchronous parallel coordinate minimization algorithm for MAP inference. Our theoretical analysis provides insights into the effect of stale updates on the convergence and speedups of this scheme. Our empirical results show the great potential of this approach, achieving linear speedups with up to 46 concurrent threads.

Future work may include improving the analysis (possibly under additional assumptions), particularly the restriction on the gradients in Theorems 1 and 2. An interesting extension of our work is to derive asynchronous parallel coordinate minimization algorithms for other objective functions, including those arising in other inference tasks, such as marginal inference. Another natural extension is to try our algorithms on MAP problems from other domains, such as natural language processing and computational Biology, adding to our experiments on disparity estimation in computer vision.

**Acknowledgments**

This material is based upon work supported in part by the National Science Foundation under Grant No. 1718221. This work utilized computing resources provided by the Innovative Systems Lab (ISL) at NCSA.

## Footnotes

[1] For a single coordinate this is equivalent to exact line search, but for larger blocks the updates can differ.

[2]The problem in Eq. (2) can also be derived as the dual of a linear programming relaxation of Eq. (1).

[3]Similar updates for the non-smooth case (Eq. (2)) are also known. Those are easily obtained by switching from soft-max to max.

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
