[Supplementary Material]

# Supplementary Material
## Asynchronous Parallel Coordinate Minimization for MAP Inference

## A    Analysis for Pencil Block

In this section we provide full derivations of the results on the Pencil block in Section 4.

### A.1    Proof of Proposition 1

*Proof.* We begin by characterizing the convexity of $f(\delta)$. For all $\delta^1, \delta^2$ we have that

$$f(\delta^2) = f(\delta^1) + \nabla f(\delta^1)^\top (\delta^2 - \delta^1) + \gamma D(\mu(\delta^1)||\mu(\delta^2)) \,,$$

where $D(p||q)$ is the KL divergence. Proof:

$$\gamma D(\mu(\delta^1)||\mu(\delta^2)) = \gamma \sum_r \sum_{x_r} \mu_r^1(x_r) \log \mu_r^1(x_r) - \gamma \sum_r \sum_{x_r} \mu_r^1(x_r) \log \mu_r^2(x_r)$$

$$= \gamma \sum_r \sum_{x_r} \mu_r^1(x_r) \left( \tilde\theta_r^{\delta^1}(x_r)/\gamma - \log \sum_{x_r'} \exp\left( \tilde\theta_r^{\delta^1}(x_r')/\gamma \right) \right)$$

$$- \gamma \sum_r \sum_{x_r} \mu_r^1(x_r) \left( \tilde\theta_r^{\delta^2}(x_r)/\gamma - \log \sum_{x_r'} \exp\left( \tilde\theta_r^{\delta^2}(x_r')/\gamma \right) \right)$$

$$= -\gamma \sum_r \log \sum_{x_r'} \exp\left( \tilde\theta_r^{\delta^1}(x_r')/\gamma \right) \overbrace{\sum_{x_r} \mu_r^1(x_r)}^{=1}$$

$$+ \gamma \sum_r \log \sum_{x_r'} \exp\left( \tilde\theta_r^{\delta^2}(x_r')/\gamma \right) \overbrace{\sum_{x_r} \mu_r^1(x_r)}^{=1}$$

$$+ \sum_r \sum_{x_r} \mu_r^1(x_r) \left( \tilde\theta_r^{\delta^1}(x_r) - \tilde\theta_r^{\delta^2}(x_r) \right)$$

$$= f(\delta^2) - f(\delta^1) + \sum_r \sum_{p:r\in p} \sum_{x_r} (\mu_r^1(x_r) - \mu_p^1(x_r))(\delta_{pr}^1(x_r) - \delta_{pr}^2(x_r))$$

$$= f(\delta^2) - f(\delta^1) + \nabla f(\delta^1)^\top (\delta^1 - \delta^2)$$

Hence, plugging $\delta^1 = \delta^t$ and $\delta^2 = \delta^{t+1}$ and taking expectation, we get:[4]

$$\mathbb{E}_s[f(\delta^{t+1})] = \mathbb{E}_s \left[ f(\delta^t) + \langle \nabla_s f(\delta^t), (\delta^{t+1} - \delta^t)_s \rangle + \gamma D(\mu(\delta^t)||\mu(\delta^{t+1})) \right]$$

$$= f(\delta^t) + \frac{1}{n} \langle \nabla f(\delta^t), \bar\delta^{t+1} - \delta^t \rangle + \gamma \mathbb{E}_s \left[ D(\mu(\delta^t)||\mu(\delta^{t+1})) \right] \qquad (8)$$

where $s$ is a randomly chosen block, $n$ is the total number of blocks, and $\bar\delta^{t+1}$ is obtained by applying the optimal update to all blocks simultaneously.

For the Pencil update we have:

$$\frac{1}{n} \langle \nabla f(\delta^t), \bar\delta^{t+1} - \delta^t \rangle = \frac{\gamma}{2n} \sum_r \sum_{p:r\in p} \sum_{x_r} (\mu_r^t(x_r) - \mu_p^t(x_r))(\log \mu_p^{k(t)}(x_r) - \log \mu_r^{k(t)}(x_r)) \qquad (9)$$

where $k(t)$ is the iteration used to compute the $t$'th update.

As for the KL term,

$$
\gamma \mathbb{E}_s \left[ D(\mu(\delta^t) || \mu(\delta^{t+1})) \right]
$$
$$
= \frac{\gamma}{n} \sum_r \sum_{p:r\in p} \sum_{r'} D(\mu_{r'}(\delta^t) || \mu_{r'}(\delta^{t+1,pr}))
$$
$$
= \frac{\gamma}{n} \sum_r \sum_{p:r\in p} \left( D(\mu_r(\delta^t) || \mu_r(\delta^{t+1,pr})) + D(\mu_p(\delta^t) || \mu_p(\delta^{t+1,pr})) \right)
$$

where the last equality holds since whenever $r' \notin \{r, p\}$, then $\mu_{r'}(\delta^t) = \mu_{r'}(\delta^{t+1,pr})$, so the KL equals 0.

Now,

$$
D(\mu_r(\delta^t) || \mu_r(\delta^{t+1,pr}))
$$
$$
= \sum_{x_r} \mu_r^t(x_r) \log \mu_r^t(x_r) - \sum_{x_r} \mu_r^t(x_r) \log \mu_r^{t+1,pr}(x_r)
$$
$$
= \sum_{x_r} \mu_r^t(x_r) \log \mu_r^t(x_r)
$$
$$
- \sum_{x_r} \mu_r^t(x_r) \left[ \frac{1}{\gamma} \left( \theta_r(x_r) + \sum_{p':r\in p'} \delta_{p'r}^{t+1,pr}(x_r) - \sum_{c':c'\in r} \delta_{rc'}^{t+1,pr}(x_c) \right) \right.
$$
$$
\left. - \log \sum_{x_r'} \exp \left( \frac{1}{\gamma} \left( \theta_r(x_r') + \sum_{p':r\in p'} \delta_{p'r}^{t+1,pr}(x_r') - \sum_{c':c'\in r} \delta_{rc'}^{t+1,pr}(x_{c'}') \right) \right) \right]
$$
$$
= \sum_{x_r} \mu_r^t(x_r) \left[ \frac{1}{\gamma} \left( \theta_r(x_r) + \sum_{p':r\in p'} \delta_{p'r}^t(x_r) - \sum_{c':c'\in r} \delta_{rc'}^t(x_c) \right) \right.
$$
$$
\left. - \log \sum_{x_r'} \exp \left( \frac{1}{\gamma} \left( \theta_r(x_r') + \sum_{p':r\in p'} \delta_{p'r}^t(x_r') - \sum_{c':c'\in r} \delta_{rc'}^t(x_{c'}') \right) \right) \right]
$$
$$
- \sum_{x_r} \mu_r^t(x_r) \left[ \frac{1}{\gamma} \left( \theta_r(x_r) + \sum_{p':r\in p'} \delta_{p'r}^t(x_r) - \sum_{c':c'\in r} \delta_{rc'}^t(x_c) + \frac{\gamma}{2}(\log \mu_p^{k(t)}(x_r) - \log \mu_r^{k(t)}(x_r)) \right) \right.
$$
$$
\left. - \log \sum_{x_r'} \exp \left( \frac{1}{\gamma} \left( \theta_r(x_r') + \sum_{p':r\in p'} \delta_{p'r}^t(x_r') - \sum_{c':c'\in r} \delta_{rc'}^t(x_{c'}') + \frac{\gamma}{2}(\log \mu_p^{k(t)}(x_r') - \log \mu_r^{k(t)}(x_r')) \right) \right) \right]
$$
$$
= -\frac{1}{2} \sum_{x_r} \mu_r^t(x_r)(\log \mu_p^{k(t)}(x_r) - \log \mu_r^{k(t)}(x_r))
$$
$$
+ \log \frac{\sum_{x_r} \exp \left( \frac{1}{\gamma} \left( \theta_r(x_r) + \sum_{p':r\in p'} \delta_{p'r}^t(x_r) - \sum_{c':c'\in r} \delta_{rc'}^t(x_{c'}) \right) \right) \cdot \exp \left( \frac{1}{2}(\log \mu_p^{k(t)}(x_r) - \log \mu_r^{k(t)}(x_r)) \right)}{\sum_{x_r'} \exp \left( \frac{1}{\gamma} \left( \theta_r(x_r') + \sum_{p':r\in p'} \delta_{p'r}^t(x_r') - \sum_{c':c'\in r} \delta_{rc'}^t(x_{c'}') \right) \right)}
$$
$$
= -\frac{1}{2} \sum_{x_r} \mu_r^t(x_r)(\log \mu_p^{k(t)}(x_r) - \log \mu_r^{k(t)}(x_r)) + \log \sum_{x_r} \mu_r^t(x_r) \sqrt{\frac{\mu_p^{k(t)}(x_r)}{\mu_r^{k(t)}(x_r)}} \quad (10)
$$

Similarly,

$$
\begin{aligned}
&D(\mu_p(\delta^t)||\mu_p(\delta^{t+1,pr})) \\
&= \sum_{x_p} \mu_p^t(x_p) \log \mu_p^t(x_p) - \sum_{x_p} \mu_p^t(x_p) \log \mu_p^{t+1,pr}(x_p) \\
&= \sum_{x_p} \mu_p^t(x_p) \log \mu_p^t(x_p) \\
&\quad - \sum_{x_p} \mu_p^t(x_p) \left[ \frac{1}{\gamma} \left( \theta_p(x_p) + \sum_{q:p\in q} \delta_{qp}^{t+1,pr}(x_p) - \sum_{r':r'\in p} \delta_{pr'}^{t+1,pr}(x_{r'}) \right) \right. \\
&\qquad \left. - \log \sum_{x_p'} \exp \left( \frac{1}{\gamma} \left( \theta_p(x_p') + \sum_{q:p\in q} \delta_{qp}^{t+1,pr}(x_p') - \sum_{r':r'\in p} \delta_{pr'}^{t+1,pr}(x_{r'}') \right) \right) \right] \\
&= \sum_{x_p} \mu_p^t(x_p) \left[ \frac{1}{\gamma} \left( \theta_p(x_p) + \sum_{q:p\in q} \delta_{qp}^{t}(x_p) - \sum_{r':r'\in p} \delta_{pr'}^{t}(x_{r'}) \right) \right. \\
&\qquad \left. - \log \sum_{x_p'} \exp \left( \frac{1}{\gamma} \left( \theta_p(x_p') + \sum_{q:p\in q} \delta_{qp}^{t}(x_p') - \sum_{r':r'\in p} \delta_{pr'}^{t}(x_{r'}') \right) \right) \right] \\
&\quad - \sum_{x_p} \mu_p^t(x_p) \left[ \frac{1}{\gamma} \left( \theta_p(x_p) + \sum_{q:p\in q} \delta_{qp}^{t}(x_p) - \sum_{r':r'\in p} \delta_{pr'}^{t}(x_{r'}) - \frac{\gamma}{2}(\log \mu_p^{k(t)}(x_r) - \log \mu_r^{k(t)}(x_r)) \right) \right. \\
&\qquad \left. - \log \sum_{x_p'} \exp \left( \frac{1}{\gamma} \left( \theta_p(x_p') + \sum_{q:p\in q} \delta_{qp}^{t}(x_p') - \sum_{r':r'\in p} \delta_{pr'}^{t}(x_{r'}') - \frac{\gamma}{2}(\log \mu_p^{k(t)}(x_r') - \log \mu_r^{k(t)}(x_r')) \right) \right) \right] \\
&= \frac{1}{2} \sum_{x_r} \mu_p^t(x_r)(\log \mu_p^{k(t)}(x_r) - \log \mu_r^{k(t)}(x_r)) \\
&\quad + \log \frac{\sum_{x_p} \exp \left( \frac{1}{\gamma} \left( \theta_p(x_p) + \sum_{q:p\in q} \delta_{qp}^{t}(x_p) - \sum_{r':r'\in p} \delta_{pr'}^{t}(x_{r'}) \right) \right) \cdot \exp \left( -\frac{1}{2}(\log \mu_p^{k(t)}(x_r) - \log \mu_r^{k(t)}(x_r)) \right)}{\sum_{x_p'} \exp \left( \frac{1}{\gamma} \left( \theta_p(x_p') + \sum_{q:p\in q} \delta_{qp}^{t}(x_p') - \sum_{r':r'\in p} \delta_{pr'}^{t}(x_{r'}') \right) \right)} \\
&= \frac{1}{2} \sum_{x_r} \mu_p^t(x_r)(\log \mu_p^{k(t)}(x_r) - \log \mu_r^{k(t)}(x_r)) + \log \sum_{x_r} \mu_p^t(x_r) \sqrt{\frac{\mu_r^{k(t)}(x_r)}{\mu_p^{k(t)}(x_r)}}
\end{aligned}
\tag{11}
$$

Combining Eq. (9), Eq. (10), and Eq. (11), we obtain:

$$
\mathbb{E}_s[f(\delta^{t+1})] = f(\delta^t) + \frac{\gamma}{n} \sum_r \sum_{p:r\in p} \left( \log \sum_{x_r} \mu_r^t(x_r) \sqrt{\frac{\mu_p^{k(t)}(x_r)}{\mu_r^{k(t)}(x_r)}} + \log \sum_{x_r} \mu_p^t(x_r) \sqrt{\frac{\mu_r^{k(t)}(x_r)}{\mu_p^{k(t)}(x_r)}} \right)
$$

Or equivalently,

$$
\begin{aligned}
\mathbb{E}_s[f(\delta^{t+1})] - f(\delta^t) = \frac{\gamma}{n} \sum_r \sum_{p:r\in p} &\left( \log \sum_{x_r} \frac{\mu_r^t(x_r)}{\mu_r^{k(t)}(x_r)} \sqrt{\mu_p^{k(t)}(x_r) \cdot \mu_r^{k(t)}(x_r)} \right. \\
&\left. + \log \sum_{x_r} \frac{\mu_p^t(x_r)}{\mu_p^{k(t)}(x_r)} \sqrt{\mu_p^{k(t)}(x_r) \cdot \mu_r^{k(t)}(x_r)} \right)
\end{aligned}
\tag{12}
$$

□

## A.2 Proof of Proposition 2

*Proof.* We first isolate the delay term from the improvement term.

$$\log \sum_{x_r} \frac{\mu_r^t(x_r)}{\mu_r^{k(t)}(x_r)} \sqrt{\mu_p^{k(t)}(x_r) \cdot \mu_r^{k(t)}(x_r)} + \log \sum_{x_r} \frac{\mu_p^t(x_r)}{\mu_p^{k(t)}(x_r)} \sqrt{\mu_p^{k(t)}(x_r) \cdot \mu_r^{k(t)}(x_r)}$$

$$\leq \log \left[ \left( \max_{x_r} \frac{\mu_r^t(x_r)}{\mu_r^{k(t)}(x_r)} \right) \left( \sum_{x_r} \sqrt{\mu_p^{k(t)}(x_r) \cdot \mu_r^{k(t)}(x_r)} \right) \right]$$

$$+ \log \left[ \left( \max_{x_r} \frac{\mu_p^t(x_r)}{\mu_p^{k(t)}(x_r)} \right) \left( \sum_{x_r} \sqrt{\mu_p^{k(t)}(x_r) \cdot \mu_r^{k(t)}(x_r)} \right) \right] \qquad \text{[Hölder]}$$

$$= \log \max_{x_r} \frac{\mu_r^t(x_r)}{\mu_r^{k(t)}(x_r)} + \log \max_{x_r} \frac{\mu_p^t(x_r)}{\mu_p^{k(t)}(x_r)} + \log \left( \sum_{x_r} \sqrt{\mu_p^{k(t)}(x_r) \cdot \mu_r^{k(t)}(x_r)} \right)^2$$

We use the monotonic increase of logarithm to get $\max \log = \log \max$.

Plugging this back into the decrease Eq. (12):

$$\mathbb{E}_s[f(\delta^{t+1})] - f(\delta^t)$$

$$\leq \frac{\gamma}{n} \sum_r \sum_{p:r\in p} \left( \max_{x_r} \log \frac{\mu_r^t(x_r)}{\mu_r^{k(t)}(x_r)} + \max_{x_r} \log \frac{\mu_p^t(x_r)}{\mu_p^{k(t)}(x_r)} \right) \qquad \text{[delay]}$$

$$+ \frac{\gamma}{n} \sum_r \sum_{p:r\in p} \log \left( \sum_{x_r} \sqrt{\mu_p^{k(t)}(x_r) \cdot \mu_r^{k(t)}(x_r)} \right)^2 \qquad \text{[stale "improvement"]} \qquad (13)$$

Next, notice that:

$$\log \frac{\mu_r^t(x_r)}{\mu_r^{k(t)}(x_r)} = \log \mu_r^t(x_r) - \log \mu_r^{k(t)}(x_r)$$

$$= \sum_{d=k(t)}^{t-1} (\log \mu_r^{d+1,pr(d)}(x_r) - \log \mu_r^d(x_r))$$

$$= \sum_{d=k(t)}^{t-1} \log \frac{\mu_r^{d+1,pr(d)}(x_r)}{\mu_r^d(x_r)}$$

where $pr(d)$ denotes the block chosen to be updated at time $d$. And likewise for $\mu_p(x_r)$.

Plugging this in the delay:

$$\frac{\gamma}{n} \sum_r \sum_{p:r\in p} \left( \max_{x_r} \log \frac{\mu_r^t(x_r)}{\mu_r^{k(t)}(x_r)} + \max_{x_r} \log \frac{\mu_p^t(x_r)}{\mu_p^{k(t)}(x_r)} \right)$$

$$= \frac{\gamma}{n} \sum_r \sum_{p:r\in p} \left[ \max_{x_r} \left( \sum_{d=k(t)}^{t-1} \log \frac{\mu_r^{d+1,pr(d)}(x_r)}{\mu_r^d(x_r)} \right) + \max_{x_r} \left( \sum_{d=k(t)}^{t-1} (\log \frac{\mu_p^{d+1,pr(d)}(x_r)}{\mu_p^d(x_r)}) \right) \right]$$

$$\leq \frac{\gamma}{n} \sum_r \sum_{p:r\in p} \left[ \sum_{d=k(t)}^{t-1} \left( \max_{x_r} \log \frac{\mu_r^{d+1,pr(d)}(x_r)}{\mu_r^d(x_r)} \right) + \sum_{d=k(t)}^{t-1} \left( \max_{x_r} \log \frac{\mu_p^{d+1,pr(d)}(x_r)}{\mu_p^d(x_r)} \right) \right]$$

$$= \frac{\gamma}{n} \sum_{d=k(t)}^{t-1} \sum_r \sum_{p:r\in p} \left[ \max_{x_r} \left( \log \frac{\mu_r^{d+1,pr(d)}(x_r)}{\mu_r^d(x_r)} \right) + \max_{x_r} \left( \log \frac{\mu_p^{d+1,pr(d)}(x_r)}{\mu_p^d(x_r)} \right) \right]$$

where the inequality follows from $(\max \sum \leq \sum \max)$.

Now notice that if we update the $pr$ block, then the only beliefs that change are $\mu_r$ and $\mu_p$. Therefore, whenever $r \notin pr(d)$ (so $r$ is neither the child nor parent in the update) then the beliefs are the same and the log terms equal 0. So we can get rid of the sums:

$$= \frac{\gamma}{n} \sum_{d=k(t)}^{t-1} \left[ \max_{x_r} \left( \log \frac{\mu_{r(d)}^{d+1,pr(d)}(x_r)}{\mu_{r(d)}^{d}(x_r)} \right) + \max_{x_r} \left( \log \frac{\mu_{p(d)}^{d+1,pr(d)}(x_r)}{\mu_{p(d)}^{d}(x_r)} \right) \right] \tag{14}$$

Finally, we obtain:

$$\mathbb{E}_s[f(\delta^{t+1})] - f(\delta^t)$$

$$\leq \frac{\gamma}{n} \sum_{d=k(t)}^{t-1} \left[ \max_{x_r} \left( \log \frac{\mu_{r(d)}^{d+1,pr(d)}(x_r)}{\mu_{r(d)}^{d}(x_r)} \right) + \max_{x_r} \left( \log \frac{\mu_{p(d)}^{d+1,pr(d)}(x_r)}{\mu_{p(d)}^{d}(x_r)} \right) \right]$$

$$+ \frac{\gamma}{n} \sum_{r} \sum_{p:r\in p} \log \left( \sum_{x_r} \sqrt{\mu_p^{k(t)}(x_r) \cdot \mu_r^{k(t)}(x_r)} \right)^2 \tag{15}$$

$\square$

## A.3  Limitation of the bound in Proposition 2

In this section we show the hardness of translating the bound in Proposition 2 to an overall rate. Let us focus on a single belief $\mu_r(x_r)$ and examine the terms in the bound which involve this belief (assuming it appears only once in the summation of Eq. (6)). We begin by expanding the expression in Eq. (6).

$$\log \frac{\mu_{r(d)}^{d+1,pr(d)}(x_r)}{\mu_{r(d)}^{d}(x_r)}$$

$$= \log \mu_{r(d)}^{d+1,pr(d)}(x_r) - \log \mu_{r(d)}^{d}(x_r)$$

$$= \log \frac{\exp\left( \frac{1}{\sigma}\hat{\theta}_{r(d)}^{\delta^d}(x_r) + \frac{1}{2}\log \frac{\mu_{p(d)}^{k(d)}(x_r)}{\mu_{r(d)}^{k(d)}(x_r)} \right)}{\sum_{x_r'} \exp\left( \frac{1}{\sigma}\hat{\theta}_{r(d)}^{\delta^d}(x_r') + \frac{1}{2}\log \frac{\mu_{p(d)}^{k(d)}(x_r')}{\mu_{r(d)}^{k(d)}(x_r')} \right)} - \log \frac{\exp\left( \frac{1}{\sigma}\hat{\theta}_{r(d)}^{\delta^d}(x_r) \right)}{\sum_{x_r''} \exp\left( \frac{1}{\sigma}\hat{\theta}_{r(d)}^{\delta^d}(x_r'') \right)}$$

$$= \frac{1}{\sigma}\hat{\theta}_{r(d)}^{\delta^d}(x_r) + \frac{1}{2}\log \frac{\mu_{p(d)}^{k(d)}(x_r)}{\mu_{r(d)}^{k(d)}(x_r)} - \log \sum_{x_r'} \exp\left( \frac{1}{\sigma}\hat{\theta}_{r(d)}^{\delta^d}(x_r') + \frac{1}{2}\log \frac{\mu_{p(d)}^{k(d)}(x_r')}{\mu_{r(d)}^{k(d)}(x_r')} \right)$$

$$- \frac{1}{\sigma}\hat{\theta}_{r(d)}^{\delta^d}(x_r) + \log \sum_{x_r''} \exp\left( \frac{1}{\sigma}\hat{\theta}_{r(d)}^{\delta^d}(x_r'') \right)$$

$$= \frac{1}{2}\log \frac{\mu_{p(d)}^{k(d)}(x_r)}{\mu_{r(d)}^{k(d)}(x_r)} - \log \frac{\sum_{x_r'} \exp\left( \hat{\theta}_{r(d)}^{\delta^d}(x_r') \right) \cdot \exp\left( \frac{1}{2}\log \frac{\mu_{p(d)}^{k(d)}(x_r')}{\mu_{r(d)}^{k(d)}(x_r')} \right)}{\sum_{x_r''} \exp\left( \hat{\theta}_{r(d)}^{\delta^d}(x_r'') \right)}$$

$$= \log \sqrt{\frac{\mu_{p(d)}^{k(d)}(x_r)}{\mu_{r(d)}^{k(d)}(x_r)}} - \log \sum_{x_r'} \mu_{r(d)}^{d}(x_r') \sqrt{\frac{\mu_{p(d)}^{k(d)}(x_r')}{\mu_{r(d)}^{k(d)}(x_r')}}$$

$$= \log \sqrt{\frac{\mu_{p(d)}^{k(d)}(x_r)}{\mu_{r(d)}^{k(d)}(x_r)}} - \log \sum_{x_r'} \frac{\mu_{r(d)}^{d}(x_r')}{\mu_{r(d)}^{k(d)}(x_r')} \sqrt{\mu_{r(d)}^{k(d)}(x_r')\mu_{p(d)}^{k(d)}(x_r')}$$

Adding the corresponding improvement term from Eq. (7),

$$\log \sqrt{\frac{\mu_{p(d)}^{k(d)}(x_r)}{\mu_{r(d)}^{k(d)}(x_r)}} - \log \sum_{x_r'} \frac{\mu_{r(d)}^{d}(x_r')}{\mu_{r(d)}^{k(d)}(x_r')} \sqrt{\mu_{r(d)}^{k(d)}(x_r')\mu_{p(d)}^{k(d)}(x_r')} + 2\log \sum_{x_r'} \sqrt{\mu_{r(d)}^{k(d)}(x_r')\mu_{p(d)}^{k(d)}(x_r')}$$

Focusing on the binary case, let $\mu_r^{k(d)}(0) = p$, $\mu_p^{k(d)}(0) = q$, and $\mu_r^d(0) = p'$. We obtain,

$$\log\sqrt{\frac{q}{p}} - \log\left(\frac{p'}{p}\sqrt{pq} + \frac{1-p'}{1-p}\sqrt{(1-p)(1-q)}\right) + 2\log\left(\sqrt{pq} + \sqrt{(1-p)(1-q)}\right).$$

Now, consider the case where $p' = p$ (old and new beliefs are equal), we obtain,

$$\log\sqrt{\frac{q}{p}} + \log\left(\sqrt{pq} + \sqrt{(1-p)(1-q)}\right)$$

Obviously, when $p \to 0$ and $q \neq 0$ this expression can grow unbounded.

## A.4 Proof of Theorem 1

*Proof.* First, since $|\hat{\theta}_r^\delta(x_r)| \leq M$, we can get a bound on the ratio $\mu_r^{d+1}(x_r)/\mu_r^d(x_r) \leq e^{2M/\gamma}$ (and likewise for $p$). Plugging this in Eq. (15), we get:

$$\mathbb{E}_s[f(\delta^{t+1})] - f(\delta^t) \leq \tau\frac{\gamma}{n}4M/\gamma + \frac{\gamma}{n}\sum_r\sum_{p:r\in p}\log\left(\sum_{x_r}\sqrt{\mu_p^{k(t)}(x_r)\cdot\mu_r^{k(t)}(x_r)}\right)^2$$

Then, since $\|\nabla f\|^2 \geq c > 0$, we have that:

$$\sum_r\sum_{p:r\in p}\log\left(\sum_{x_r}\sqrt{\mu_p^{k(t)}(x_r)\cdot\mu_r^{k(t)}(x_r)}\right)^2 \leq -\frac{1}{4}\sum_r\sum_{p:r\in p}\sum_{x_r}\left(\mu_p^{k(t)}(x_r) - \mu_r^{k(t)}(x_r)\right)^2 \leq -\frac{c}{4}$$

(this is a known inequality from information theory – see Proposition 1.4 in Meshi et al. [2014]).

We get that for any $\tau \leq \frac{(1-\rho)\gamma c}{16M}$, it holds that:

$$\begin{aligned}
\mathbb{E}_s[f(\delta^{t+1})] - f(\delta^t) &\leq \frac{(1-\rho)\gamma c}{16M}\frac{\gamma}{n}4M/\gamma + \frac{\gamma}{n}\sum_r\sum_{p:r\in p}\log\left(\sum_{x_r}\sqrt{\mu_p^{k(t)}(x_r)\cdot\mu_r^{k(t)}(x_r)}\right)^2 \\
&\leq -(1-\rho)\frac{\gamma}{n}\sum_r\sum_{p:r\in p}\log\left(\sum_{x_r}\sqrt{\mu_p^{k(t)}(x_r)\cdot\mu_r^{k(t)}(x_r)}\right)^2 \\
&\quad + \frac{\gamma}{n}\sum_r\sum_{p:r\in p}\log\left(\sum_{x_r}\sqrt{\mu_p^{k(t)}(x_r)\cdot\mu_r^{k(t)}(x_r)}\right)^2 \\
&= \rho\frac{\gamma}{n}\sum_r\sum_{p:r\in p}\log\left(\sum_{x_r}\sqrt{\mu_p^{k(t)}(x_r)\cdot\mu_r^{k(t)}(x_r)}\right)^2
\end{aligned}$$

Which gives the desired expected decrease.
Following the derivation in Theorem 1 of Nesterov [2012], the expected decrease can be translated into convergence rate:

$$\mathbb{E}[f(\delta^t)] - f(\delta^*) \leq \frac{4nB}{\rho\gamma t}$$

where $B$ is a scalar such that $\|\delta^t - \delta^*\|^2 \leq B$.
Setting $\rho = 1/2$ completes the proof. $\qquad\square$

# B Analysis for Star Block

In this section we present analysis for the Star block, analogous to the results on Pencil block in Section 4. Detailed proofs are given below.

**Proposition 3.** *The APCM-Star algorithm satisfies:*

$$\mathbb{E}_s[f(\delta^{t+1})] - f(\delta^t) = \frac{\gamma}{n}\sum_r\left[\log\sum_{x_r}\frac{\mu_r^t(x_r)}{\mu_r^{k(t)}(x_r)}\left(\mu_r^{k(t)}(x_r)\prod_{p:r\in p}\mu_p^{k(t)}(x_r)\right)^{\frac{1}{P_r+1}}\right. \tag{16}$$

$$\left. + \sum_{p:r\in p}\log\sum_{x_r}\frac{\mu_p^t(x_r)}{\mu_p^{k(t)}(x_r)}\left(\mu_r^{k(t)}(x_r)\prod_{p':r\in p'}\mu_{p'}^{k(t)}(x_r)\right)^{\frac{1}{P_r+1}}\right],$$

*where $n = |\mathcal{R}|$ is the number of Star blocks (slightly overloading notation).*

**Proposition 4.** *The APCM-Star algorithm satisfies:*

$$\mathbb{E}_s[f(\delta^{t+1})] - f(\delta^t) \leq \frac{\gamma}{n} \sum_{d=k(t)}^{t-1} \left[ \max_{x_r} \left( \log \frac{\mu_{r(d)}^{d+1}(x_r)}{\mu_{r(d)}^d(x_r)} \right) + \sum_{p:r(d)\in p} \max_{x_r} \left( \log \frac{\mu_p^{d+1}(x_r)}{\mu_p^d(x_r)} \right) \right] \qquad (17)$$

$$+ \frac{\gamma}{n} \sum_r \log \left( \sum_{x_r} \left( \mu_r^{k(t)}(x_r) \prod_{p':r\in p'} \mu_{p'}^{k(t)}(x_r) \right)^{\frac{1}{P_r+1}} \right)^{P_r+1} .$$

As in the case of the Pencil block, whenever there is no delay ($\tau = 0$), we recover the sequential expected decrease in objective. Here this corresponds to the (negative) Matusita divergence measure, which generalizes the Bhattacharyya divergence to multiple distributions [Meshi et al., 2014].

**Theorem 2.** *Let $|\hat{\theta}_r^{\delta^t}(x_r)| \leq M$ for all $t, r, x_r$, and let $\|\delta^t - \delta^*\|^2 < B$ for all $t$. Assume that the gradient is bounded from below as $\|\nabla f\|^2 \geq c$, and that the delay is bounded as $\tau \leq \frac{\gamma c}{16\bar{P}(\bar{P}+1)M}$, where $\bar{P} = \max_r P_r$. Then $\mathbb{E}_s[f(\delta^t)] - f(\delta^*) \leq \frac{8n\bar{P}B}{\gamma t}$.*

As in Theorem 1, this rate is 2 times slower than the sequential rate in terms of the number of iterations, but we can execute on the order of $\tau$ times more iterations at the same time, obtaining a linear speedup. Also notice that the assumption on the delay has inverse dependence on the number of parents in the region graph ($\bar{P}$). If the graph is densely connected, our theory suggests we cannot afford a very large delay.

## B.1 Proof of Proposition 3

*Proof.* For the Star block, we have:

$$\frac{1}{n} \langle \nabla f(\delta^t), \bar{\delta}^{t+1} - \delta^t \rangle$$

$$= \frac{\gamma}{n} \sum_r \sum_{p:r\in p} \sum_{x_r} (\mu_r^t(x_r) - \mu_p^t(x_r)) \left( \log \mu_p^{k(t)}(x_r) - \frac{1}{P_r+1} \left( \log \mu_r^{k(t)}(x_r) + \sum_{p':r\in p'} \log \mu_{p'}^{k(t)}(x_r) \right) \right)$$

For the KL term we have:

$$\gamma \mathbb{E}_s \left[ D(\mu(\delta^t) || \mu(\delta^{t+1})) \right]$$

$$= \frac{\gamma}{n} \sum_r \sum_{r'} D(\mu_{r'}(\delta^t) || \mu_{r'}(\delta^{t+1,r}))$$

$$= \frac{\gamma}{n} \sum_r \left( D(\mu_r(\delta^t) || \mu_r(\delta^{t+1,r})) + \sum_{p:r\in p} D(\mu_p(\delta^t) || \mu_p(\delta^{t+1,r})) \right)$$

Now,

$$D(\mu_r(\delta^t)||\mu_r(\delta^{t+1,r}))$$

$$= \sum_{x_r} \mu_r^t(x_r) \log \mu_r^t(x_r) - \sum_{x_r} \mu_r^t(x_r) \log \mu_r^{t+1,r}(x_r)$$

$$= \sum_{x_r} \mu_r^t(x_r) \log \mu_r^t(x_r)$$

$$- \sum_{x_r} \mu_r^t(x_r) \left[ \frac{1}{\gamma} \left( \theta_r(x_r) + \sum_{p:r\in p} \delta_{pr}^{t+1,r}(x_r) - \sum_{c:c\in r} \delta_{rc}^{t+1,r}(x_c) \right) \right.$$

$$\left. - \log \sum_{x_r'} \exp \left( \frac{1}{\gamma} \left( \theta_r(x_r') + \sum_{p:r\in p} \delta_{pr}^{t+1,r}(x_r') - \sum_{c:c\in r} \delta_{rc}^{t+1,r}(x_c') \right) \right) \right]$$

$$= \sum_{x_r} \mu_r^t(x_r) \left[ \frac{1}{\gamma} \left( \theta_r(x_r) - \sum_{c:c\in r} \delta_{rc}^t(x_c) + \sum_{p:r\in p} \delta_{pr}^t(x_r) \right) \right.$$

$$\left. - \log \sum_{x_r'} \exp \left( \frac{1}{\gamma} \left( \theta_r(x_r') + \sum_{p:r\in p} \delta_{pr}^t(x_r') - \sum_{c:c\in r} \delta_{rc}^t(x_c') \right) \right) \right]$$

$$- \sum_{x_r} \mu_r^t(x_r) \left[ \left( \frac{1}{\gamma} \left( \theta_r(x_r) - \sum_{c:c\in r} \delta_{rc}^t(x_c) + \sum_{p:r\in p} \delta_{pr}^t(x_r) + \gamma \log \mu_p^{k(t)}(x_r) - \frac{\gamma}{P_r + 1} \left( \log \mu_r^{k(t)}(x_r) + \sum_{p':r\in p'} \log \mu_{p'}^{k(t)}(x_r) \right) \right) \right) \right.$$

$$\left. - \log \sum_{x_r'} \exp \left( \frac{1}{\gamma} \left( \theta_r(x_r') - \sum_{c:c\in r} \delta_{rc}^t(x_c') + \sum_{p:r\in p} \delta_{pr}^t(x_r') + \gamma \log \mu_p^{k(t)}(x_r') - \frac{\gamma}{P_r + 1} \left( \log \mu_r^{k(t)}(x_r') + \sum_{p':r\in p'} \log \mu_{p'}^{k(t)}(x_r') \right) \right) \right) \right]$$

$$= - \sum_{x_r} \mu_r^t(x_r) \sum_{p:r\in p} \left( \log \mu_p^{k(t)}(x_r) - \frac{1}{P_r + 1} \left( \log \mu_r^{k(t)}(x_r) + \sum_{p':r\in p'} \log \mu_{p'}^{k(t)}(x_r) \right) \right)$$

$$+ \log \sum_{x_r'} \mu_r^t(x_r') \exp \left( \sum_{p:r\in p} \log \mu_p^{k(t)}(x_r') - \frac{P_r}{P_r + 1} \left( \log \mu_r^{k(t)}(x_r') + \sum_{p':r\in p'} \log \mu_{p'}^{k(t)}(x_r') \right) \right)$$

$$= - \sum_{x_r} \mu_r^t(x_r) \sum_{p:r\in p} \log \mu_p^{k(t)}(x_r) - \frac{1}{P_r + 1} \left( \log \mu_r^{k(t)}(x_r) + \sum_{p':r\in p'} \log \mu_{p'}^{k(t)}(x_r) \right)$$

$$+ \log \sum_{x_r} \mu_r^t(x_r) \frac{\prod_{p:r\in p} \mu_p^{k(t)}(x_r)}{\left( \mu_r^{k(t)}(x_r) \prod_{p:r\in p} \mu_p^{k(t)}(x_r) \right)^{\frac{P_r}{P_r + 1}}}$$

Similarly,

$$D(\mu_p(\delta^t)||\mu_p(\delta^{t+1,r}))$$

$$= \sum_{x_p} \mu_p^t(x_p) \log \mu_p^t(x_p) - \sum_{x_p} \mu_r^t(x_p) \log \mu_p^{t+1,r}(x_p)$$

$$= \sum_{x_p} \mu_p^t(x_p) \log \mu_p^t(x_p)$$

$$- \sum_{x_p} \mu_p^t(x_p) \left[ \frac{1}{\gamma} \left( \theta_p(x_p) + \sum_{q:p\in q} \delta_{qp}^{t+1,r}(x_p) - \sum_{r':r'\in p} \delta_{pr'}^{t+1,r}(x_{r'}) \right) \right.$$

$$\left. - \log \sum_{x_p'} \exp \left( \frac{1}{\gamma} \left( \theta_p(x_p') + \sum_{q:p\in q} \delta_{qp}^{t+1,r}(x_p') - \sum_{r':r'\in p} \delta_{pr'}^{t+1,r}(x_{r'}') \right) \right) \right]$$

$$= \sum_{x_p} \mu_p^t(x_p) \left[ \frac{1}{\gamma} \left( \theta_p(x_p) + \sum_{q:p\in q} \delta_{qp}^t(x_p) - \sum_{r':r'\in p} \delta_{pr'}^t(x_{r'}) \right) \right.$$

$$\left. - \log \sum_{x_p'} \exp \left( \frac{1}{\gamma} \left( \theta_p(x_p') + \sum_{q:p\in q} \delta_{qp}^t(x_p') - \sum_{r':r'\in p} \delta_{pr'}^t(x_{r'}') \right) \right) \right]$$

$$- \sum_{x_p} \mu_p^t(x_p) \left[ \frac{1}{\gamma} \left( \theta_p(x_p) + \sum_{q:p\in q} \delta_{qp}^t(x_p) - \sum_{r':r'\in p} \delta_{pr'}^t(x_{r'}) - \left( \gamma \log \mu_p^{k(t)}(x_r) - \frac{\gamma}{P_r+1} \left( \log \mu_r^{k(t)}(x_r) + \sum_{p':r\in p'} \log \mu_{p'}^{k(t)}(x_r) \right) \right) \right) \right.$$

$$\left. - \log \sum_{x_p'} \exp \left( \frac{1}{\gamma} \left( \theta_p(x_p') + \sum_{q:p\in q} \delta_{qp}^t(x_p') - \sum_{r':r'\in p} \delta_{pr'}^t(x_{r'}') - \left( \gamma \log \mu_p^{k(t)}(x_r) - \frac{\gamma}{P_r+1} \left( \log \mu_r^{k(t)}(x_r) + \sum_{p':r\in p'} \log \mu_{p'}^{k(t)}(x_r) \right) \right) \right) \right) \right)$$

$$= \sum_{x_p} \mu_p^t(x_p) \left( \log \mu_p^{k(t)}(x_r) - \frac{1}{P_r+1} \left( \log \mu_r^{k(t)}(x_r) + \sum_{p':r\in p'} \log \mu_{p'}^{k(t)}(x_r) \right) \right)$$

$$+ \log \sum_{x_p} \mu_p^t(x_p) \exp \left( - \left( \log \mu_p^{k(t)}(x_r) - \frac{1}{P_r+1} \left( \log \mu_r^{k(t)}(x_r) + \sum_{p':r\in p'} \log \mu_{p'}^{k(t)}(x_r) \right) \right) \right)$$

$$= \sum_{x_r} \mu_p^t(x_r) \left( \log \mu_p^{k(t)}(x_r) - \frac{1}{P_r+1} \left( \log \mu_r^{k(t)}(x_r) + \sum_{p':r\in p'} \log \mu_{p'}^{k(t)}(x_r) \right) \right)$$

$$+ \log \sum_{x_r} \mu_p^t(x_r) \frac{\left( \mu_r^{k(t)}(x_r) \prod_{p':r\in p'} \mu_{p'}^{k(t)}(x_r) \right)^{\frac{1}{P_r+1}}}{\mu_p^{k(t)}(x_r)}$$

Combining everything, we get:

$$\frac{1}{n} \langle \nabla f(\delta^t), \bar{\delta}^{t+1} - \delta^t \rangle + \gamma \mathbb{E}_s \left[ D(\mu(\delta^t)||\mu(\delta^{t+1})) \right]$$

$$= \frac{\gamma}{n} \sum_r \sum_{p:r\in p} \sum_{x_r} (\mu_r^t(x_r) - \mu_p^t(x_r)) \left( \log \mu_p^{k(t)}(x_r) - \frac{1}{P_r+1} \left( \log \mu_r^{k(t)}(x_r) + \sum_{p':r\in p'} \log \mu_{p'}^{k(t)}(x_r) \right) \right)$$

$$+ \frac{\gamma}{n} \sum_r \left[ - \sum_{x_r} \mu_r^t(x_r) \sum_{p:r\in p} \log \mu_p^{k(t)}(x_r) - \frac{1}{P_r+1} \left( \log \mu_r^{k(t)}(x_r) + \sum_{p':r\in p'} \log \mu_{p'}^{k(t)}(x_r) \right) \right.$$

$$\left. + \log \sum_{x_r} \mu_r^t(x_r) \frac{\prod_{p:r\in p} \mu_p^{k(t)}(x_r)}{\left( \mu_r^{k(t)}(x_r) \prod_{p:r\in p} \mu_p^{k(t)}(x_r) \right)^{\frac{P_r}{P_r+1}}} \right]$$

$$+ \frac{\gamma}{n} \sum_r \sum_{p:r\in p} \left[ \sum_{x_r} \mu_p^t(x_r) \left( \log \mu_p^{k(t)}(x_r) - \frac{1}{P_r+1} \left( \log \mu_r^{k(t)}(x_r) + \sum_{p':r\in p'} \log \mu_{p'}^{k(t)}(x_r) \right) \right) \right.$$

$$\left. + \log \sum_{x_r} \mu_p^t(x_r) \frac{\left( \mu_r^{k(t)}(x_r) \prod_{p':r\in p'} \mu_{p'}^{k(t)}(x_r) \right)^{\frac{1}{P_r+1}}}{\mu_p^{k(t)}(x_r)} \right]$$

$$= \frac{\gamma}{n} \sum_r \left[ \log \sum_{x_r} \mu_r^t(x_r) \frac{\prod_{p:r\in p} \mu_p^{k(t)}(x_r)}{\left( \mu_r^{k(t)}(x_r) \prod_{p:r\in p} \mu_p^{k(t)}(x_r) \right)^{\frac{P_r}{P_r+1}}} \right.$$

$$\left. + \sum_{p:r\in p} \log \sum_{x_r} \mu_p^t(x_r) \frac{\left( \mu_r^{k(t)}(x_r) \prod_{p':r\in p'} \mu_{p'}^{k(t)}(x_r) \right)^{\frac{1}{P_r+1}}}{\mu_p^{k(t)}(x_r)} \right]$$

So finally,

$$\mathbb{E}_s[f(\delta^{t+1})] - f(\delta^t) = \frac{\gamma}{n} \sum_r \left[ \log \sum_{x_r} \frac{\mu_r^t(x_r)}{\mu_r^{k(t)}(x_r)} \left( \mu_r^{k(t)}(x_r) \prod_{p:r\in p} \mu_p^{k(t)}(x_r) \right)^{\frac{1}{P_r+1}} \right.$$

$$\left. + \sum_{p:r\in p} \log \sum_{x_r} \frac{\mu_p^t(x_r)}{\mu_p^{k(t)}(x_r)} \left( \mu_r^{k(t)}(x_r) \prod_{p':r\in p'} \mu_{p'}^{k(t)}(x_r) \right)^{\frac{1}{P_r+1}} \right] \quad (18)$$

$\square$

## B.2 Proof of Proposition 4

*Proof.* As in the Pencil block, we begin with separating the delay and improvement terms:

$$\log \sum_{x_r} \frac{\mu_r^t(x_r)}{\mu_r^{k(t)}(x_r)} \left( \mu_r^{k(t)}(x_r) \prod_{p:r\in p} \mu_p^{k(t)}(x_r) \right)^{\frac{1}{P_r+1}}$$

$$+ \sum_{p:r\in p} \log \sum_{x_r} \frac{\mu_p^t(x_r)}{\mu_p^{k(t)}(x_r)} \left( \mu_r^{k(t)}(x_r) \prod_{p':r\in p'} \mu_{p'}^{k(t)}(x_r) \right)^{\frac{1}{P_r+1}}$$

$$\leq \log \left[ \left( \max_{x_r} \frac{\mu_r^t(x_r)}{\mu_r^{k(t)}(x_r)} \right) \left( \sum_{x_r} \left( \mu_r^{k(t)}(x_r) \prod_{p:r\in p} \mu_p^{k(t)}(x_r) \right)^{\frac{1}{P_r+1}} \right) \right]$$

$$+ \sum_{p:r\in p} \log \left[ \left( \max_{x_r} \frac{\mu_p^t(x_r)}{\mu_p^{k(t)}(x_r)} \right) \left( \sum_{x_r} \left( \mu_r^{k(t)}(x_r) \prod_{p':r\in p'} \mu_{p'}^{k(t)}(x_r) \right)^{\frac{1}{P_r+1}} \right) \right] \qquad \text{[Hölder]}$$

$$= \max_{x_r} \log \frac{\mu_r^t(x_r)}{\mu_r^{k(t)}(x_r)} + \sum_{p:r\in p} \max_{x_r} \log \frac{\mu_p^t(x_r)}{\mu_p^{k(t)}(x_r)} \qquad \text{[delay]}$$

$$+ \log \left( \sum_{x_r} \left( \mu_r^{k(t)}(x_r) \prod_{p':r\in p'} \mu_{p'}^{k(t)}(x_r) \right)^{\frac{1}{P_r+1}} \right)^{P_r+1} \qquad \text{[stale improvement]}$$

So finally, we obtain:

$$\mathbb{E}_s[f(\delta^{t+1})] - f(\delta^t) \leq \frac{\gamma}{n} \sum_{d=k(t)}^{t-1} \left[ \max_{x_r} \left( \log \frac{\mu_{r(d)}^{d+1,r(d)}(x_r)}{\mu_{r(d)}^d(x_r)} \right) + \sum_{p:r(d)\in p} \max_{x_r} \left( \log \frac{\mu_p^{d+1,r(d)}(x_r)}{\mu_p^d(x_r)} \right) \right]$$

$$+ \frac{\gamma}{n} \sum_r \log \left( \sum_{x_r} \left( \mu_r^{k(t)}(x_r) \prod_{p':r\in p'} \mu_{p'}^{k(t)}(x_r) \right)^{\frac{1}{P_r+1}} \right)^{P_r+1}$$

$\square$

## B.3 Proof of Theorem 2

*Proof.* As in the Pencil block, we first bound the delay term using $\mu_r^{d+1}(x_r)/\mu_r^d(x_r) \leq e^{2M/\gamma}$ (and likewise for all parents $p$).

$$\mathbb{E}_s[f(\delta^{t+1})] - f(\delta^t) \leq \tau \frac{\gamma}{n} 2(\bar{P}+1)M/\gamma + \frac{\gamma}{n} \sum_r \log \left( \sum_{x_r} \left( \mu_r^{k(t)}(x_r) \prod_{p':r\in p'} \mu_{p'}^{k(t)}(x_r) \right)^{\frac{1}{P_r+1}} \right)^{P_r+1}$$

Now, using $\|\nabla f\|^2 \geq c$ and Proposition 1.5 in Meshi et al. [2014], we obtain:

$$\sum_r \log \left( \sum_{x_r} \left( \mu_r^{k(t)}(x_r) \prod_{p':r \in p'} \mu_{p'}^{k(t)}(x_r) \right)^{\frac{1}{P_r+1}} \right)^{P_r+1} \leq -\sum_r \frac{1}{4P_r} \sum_{p:r \in p} \sum_{x_r} \left( \mu_p^{k(t)}(x_r) - \mu_r^{k(t)}(x_r) \right)^2 \leq -\frac{c}{4\bar{P}}$$

Using the assumed bound on the delay $\tau \leq \frac{(1-\rho)\gamma c}{8\bar{P}(\bar{P}+1)M}$,

$$\mathbb{E}_s[f(\delta^{t+1})] - f(\delta^t) \leq \frac{(1-\rho)\gamma c}{8\bar{P}(\bar{P}+1)M} \frac{\gamma}{n}(\bar{P}+1)2M/\gamma + \frac{\gamma}{n}\sum_r \log \left( \sum_{x_r} \left( \mu_r^{k(t)}(x_r) \prod_{p':r \in p'} \mu_{p'}^{k(t)}(x_r) \right)^{\frac{1}{P_r+1}} \right)^{P_r+1}$$

$$\leq -(1-\rho)\frac{\gamma}{n}\sum_r \log \left( \sum_{x_r} \left( \mu_r^{k(t)}(x_r) \prod_{p':r \in p'} \mu_{p'}^{k(t)}(x_r) \right)^{\frac{1}{P_r+1}} \right)^{P_r+1}$$

$$+ \frac{\gamma}{n}\sum_r \log \left( \sum_{x_r} \left( \mu_r^{k(t)}(x_r) \prod_{p':r \in p'} \mu_{p'}^{k(t)}(x_r) \right)^{\frac{1}{P_r+1}} \right)^{P_r+1}$$

$$= \rho\frac{\gamma}{n}\sum_r \log \left( \sum_{x_r} \left( \mu_r^{k(t)}(x_r) \prod_{p':r \in p'} \mu_{p'}^{k(t)}(x_r) \right)^{\frac{1}{P_r+1}} \right)^{P_r+1} .$$

Finally, following the sequential analysis [Nesterov, 2012, Meshi et al., 2014] yields the rate,

$$\mathbb{E}_s[f(\delta^t)] - f(\delta^*) \leq \frac{8n\bar{P}B}{\rho\gamma t} .$$

Setting $\rho = 1/2$ completes the proof.

$\square$

## C  Additional Results

In Fig. 4 and Fig. 5 we illustrate the convergence behavior of our approach for $\gamma = 0$ and a state space size of 8, as well as for $\gamma = 1$ and state space size of 16, respectively. As before, we observe fast convergence of the parallelized approach. For both settings we illustrate in Fig. 6 the speedup w.r.t. a single thread obtained for a specific number of threads of our approach (see Fig. 6 (a)) and for HOGWILD! (see Fig. 6 (b)). The results follow the observation reported in the main paper. As observed in Fig. 6 (c), the speedups of our approach compared to HOGWILD! tend to be slightly bigger than the ones reported in the paper when considering $\gamma = 0$ or larger MRFs.

Figure 4: For $\gamma = 0$ and an 8 state model, we illustrate the convergence behavior of our approach compared to HOGWILD!, for a variety of MRF configurations (2, 4, 8), and different number of iterations (200, 400). Different number of threads are used for each configuration.

Figure 5: For $\gamma = 1$ and a 16 state model, we illustrate the convergence behavior of our approach compared to HOGWILD!, for a variety of MRF configurations (2, 4, 8), and different number of iterations (200, 400). Different number of threads are used for each configuration.

Figure 6: Speedup w.r.t. single thread obtained for a specific number of threads for our approach (a) and HOGWILD! (b), using a variety of MRF neighborhoods (2, 4, 8), and different number of iterations (200, 400). Speedups are shown for: (top) $\gamma = 1$ and 8 states, (bottom) $\gamma = 1$ and 16 states. (c) shows the speedup of our method compared to HOGWILD!.

## Footnotes

[4]The second equality assumes that that all blocks are equally likely to be picked. This may not be true, for example if some blocks take much longer to process than others (since those will be updated less frequently). However, it is a standard assumption in the existing literature.