[Reviews · NeurIPS 2017]

Reviewer 1



Summary: This paper proposes an asynchronous parallel approximate algorithm for MAP inference in graphical models represented as factor graphs. The proposed method is based on dual decomposition which breaks the model into overlapping pieces and adds penalty terms to enforce agreement between the overlapping portions. Whereas, HOGWILD performs asynchronous gradient updates at each factor, the proposed method performs full coordinate ascent at each iteration. The main concern is that updates based on stale values will be invalid, however, the authors show results that bound expected errors of this type. The authors also provide some methods for adaptively choosing the number of worker nodes to further minimize this error. Overall, I found the paper reasonably clear and the methods well justified. Conceptually, the difference between the proposed method and HOGWILD is relatively small, but the analysis is well done and the experiments (which I found well constructed and convincing) demonstrate that this change can lead to large performance gains. Comments: - I found portions of the notation to be a bit unclear. I think that the authors could spend a bit more time introducing and describing equation (2) which would seem to be the crux of the method. I suggest adding a few steps of the derivation of (2) as it is not that difficult to show and would make the paper much more accessible. Barring that, explaining what is meant by a "containment relationship" and the intuition behind the Lagrange multipliers (which are just called "agreement constraints") would help. - How is the MAP assignment recovered from a smoothed objective? (I would add this step to the text as it is not obvious) - The authors should either show the proof that the updates in (4) are the coordinate maximizers or cite the specific source that proves this. - I would consider making it more clear that the error considered in section 4 is the only type of possible error accrued by the parallelism. A simple statement such as "If there is no delay, then the algorithm is performing exact coordinate ascent" near the beginning of this section would clear this up.

Reviewer 2



SUMMARY: ======== The authors propose a parallel MAP inference algorithm based on a dual decomposition of the MAP objective. Updates are performed without synchronization and locally enforce consistency between overlapping regions. A convergence analysis establishes a linear rate of convergence and experiments are performed on synthetic MRFs and image disparity estimation. PROS: ===== The paper is well-rounded and provides algorithmic development, theoretical analysis in terms of convergence rate results, and experimental validation. Clarity is good overall and the work is interesting, novel, and well-motivated. CONS: ===== Experimental validation in the current paper is inadequate in several respects. First, since the approach is posed as a general MAP inference it would be ideal to see validation in more than one synthetic and one real world model (modulo variations in network topology). Second, the size of problem instances is much too small to validate a "large-scale" inference algorithm. Indeed, as the authors note (L:20-21) a motivation of the proposed approach is that "modern applications give rise to very large instances". To be clear, the "size" of instances in this setting is the number of regions in Eq. (1). The largest example is on image disparity, which contains less than 27K regions. By contrast, the main comparison, HOGWILD, was originally validated on three standard "real world" instances with a number of objective terms reaching into the millions. Note that even single-threaded inference converges within a few hundred seconds in most cases. Finally, the authors do not state which disparity estimation dataset and model are used. This reviewer assumes the standard Middlebury dataset it used, but the reported number of regions is much smaller than the typical image size in that dataset, please clarify. Results of the convergence analysis are not sufficient to justify the lack of experimental validation. In particular, Theorem 1 provides a linear rate of convergence, but this does not establish global linear rate of convergence as the bound is only valid away from a stationary point (e.g. when the gradient norm is above threshold). It is also unclear to what extent assumptions of the convergence rate can be evaluated since they must be such that the delay rate, which is unknown and not controlled, is bounded above as a function of c and M. As a secondary issue, this reviewer is unclear about the sequential results. Specifically, it is not clear why the upper bound in Prop. 2 is necessary when an exact calculation in Prop. 1 has been established, and as the authors point out this bound can be quite loose. Also, it is not clear what the authors mean when they state Prop. 2 accounts for the sparsity of updates. Please clarify.